

# Influence of honey bee (*Apis mellifera*) breeding on wing venation in Serbia and neighbouring countries

Hardeep Kaur[1], Nebojša Nedić[2] and Adam Tofilski[1]

[1] Department of Zoology and Animal Welfare, University of Agriculture in Krakow, Krakow, Poland
[2] Institute for Zootechnics, Faculty of Agriculture, University of Belgrade, Belgrade, Serbia

## ABSTRACT

In order to improve the productivity of honey bees (*Apis mellifera*), some of their traits are selected by breeding. On one hand, breeding is mainly based on the natural geographical variation of this species; on the other hand, mass production and distribution of artificially selected queens can significantly affect the natural geographic variation of honey bees. In this study, we have compared honey bee wings originating from breeding and non-breeding populations in Serbia. In the comparison, we have also used data from a large area of south-eastern Europe. The wings were measured using the 19 landmarks indicated on the wing images. The coordinates were analysed using the methodology of geometric morphometrics. We found that honey bees obtained from honey bee queen breeder differed in wing venation from surrounding populations, which are under natural selection. Therefore, we argue against including populations under artificial selection in the analysis of the natural geographical variation of honey bees. In our analysis of non-breeding samples, we found that in south-eastern Europe there is continuous variation in wing venation and no clear boundaries between *A. m. carnica*, *A. m. cecropia*, and *A. m. macedonica*.

# INTRODUCTION

The honey bee (*Apis mellifera*) is an important pollinator in both natural (*Hung et al., 2018*) and agricultural ecosystems (*Aizen & Harder, 2009*). Additionally, beekeeping is a significant area of agriculture since it produces honey and other products from bees (*Jones, 2004*). The productivity of the bees can be improved by breeding and artificial selection of selected traits. To do this, a group of colonies is scored according to desirable traits, and only some colonies with the highest score are selected for the next generation (*Uzunov, Brascamp & Büchler, 2017*). Repetition of this procedure over numerous generations frequently results in the improvement of some traits (*Bienefeld, 2016*). This process can be faster if molecular markers are used during the selection (*Niño & Cameron Jasper, 2015*). Control of the paternal side during honey bee breeding is difficult because mating occurs in flight at a distance from the nest (*Gries & Koeniger, 1996*). In order to control mating between selected parents, they can be isolated at a distance from other colonies. This is

Corresponding author
Adam Tofilski, rotofils@cyf-kr.edu.pl

challenging, though, because there aren't many locations in Europe that are more than 15 km apart from an apiary (*Jensen et al., 2005*). Even if in a particular area there are no colonies maintained by beekeepers in hives, a small feral population can still be present (*Kohl & Rutschmann, 2018*; *Patenković et al., 2022*). In this situation, many breeders use instrumental insemination, which requires sophisticated equipment and qualified personnel.

Before honey bee colonies were dispersed by humans all over the world, they were distributed in Europe, Africa, the Middle East, and Central Asia. Within this wide distribution, there were numerous subspecies described (*Ruttner, 1988a*), which are divided into four to seven evolutionary lineages (*Ruttner, 1988a*; *Franck et al., 2001*; *Dogantzis et al., 2021*). The subspecies differ in various morphological and behavioural traits, which reflect their adaptation to the wide range of climates in which they occur. On one hand, honey bee breeders employ this natural genetic variation to select certain features that are desired in beekeeping. On the other hand, the mass production and distribution of artificially selected queens affects markedly the natural geographic variation of honey bees (*De la Rúa et al., 2009*). The most detrimental to honey bee biodiversity is the importation of subspecies that naturally occur in different areas. In Europe, beekeepers often prefer *A. m. carnica* and *A. m. ligustica* and introduce them in the distribution range of *A. m. mellifera* (*Henriques et al., 2019*). In some cases, breeding is based on hybrids between honey bee subspecies (*Kulincevic & Rinderer, 1986*), which may contain African genetic material (*Oleksa, Kusza & Tofilski, 2021*). The introduction of non-native bees affects the entire population because beekeepers do not control mating, and alien bees introduced by one beekeeper hybridise with all surrounding colonies, both managed and feral. This process of admixture is gradual because beekeepers usually introduce un-inseminated queens, which mate with local drones. Even if the breeding is based on the native gene pool, it can reduce the genetic variability of the population because the number of colonies used in the breeding programmes is often relatively small. This problem is particularly acute if breeding is monopolised by a small number of large breeders or if the majority of breeders use a limited number of breeding lines (*Cobey, Sheppard & Tarpy, 2012*). Although breeding activities often negatively affect the biodiversity of honey bees, they can be used for their conservation (*Lodesani & Costa, 2003*; *Meixner et al., 2010*; *Oleksa et al., 2011*; *Muñoz et al., 2014*; *Requier et al., 2019*).

Honey bees in Serbia belong to the East Mediterranean mitochondrial line C (*Kozmus et al., 2007*; *Nedić et al., 2009*). In fact, Serbia is at the centre of this lineage distribution. According to *Ruttner (1988a)*, in Serbia, there were two subspecies: *A. m. carnica* in most of the country and *A. m. macedonica* in its south-eastern limits (*Ruttner, 1988a*, Fig. 14.1). This pattern was largely confirmed by some other studies (*Stevanovic et al., 2010*; *Nedić et al., 2014*). However, in some recent studies, the presence of *A. m. macedonica* in southern Serbia has not been confirmed (*Tanasković et al., 2021*, *2022*). This can be related to introgression between native and non-native bees, which was reported in Serbia (*Muñoz et al., 2012*; *Nedić et al., 2014*) and neighbouring North Macedonia (*Uzunov et al., 2009*). In countries neighbouring Serbia, there is *A. m. carnica* in Hungary (*Péntek-Zakar et al., 2015*), Croatia (*Puškadija et al., 2021*), Bosnia and Herzegovina (*Stevanovic et al., 2010*),

and Montenegro (*Rašić, Mladenović & Stanisavljević, 2015*). In Romania, there are two subpopulations on both sides of the Carpathian Mountains (*Tofilski et al., 2021*). According to *Ruttner (1988a)*, western Romania belongs to *A. m. carnica*; however, earlier, another subspecies, *A. m. carpatica*, was described there (*Foti et al., 1965*). *A. m. macedonica* was reported in North Macedonia (*Stevanovic et al., 2010*) and Bulgaria (*Ivanova, Bienkowska & Petrov, 2011*). The population from Bulgaria is sometimes considered a separate subspecies, *A. m. rodopica* (*Petrov, 1995*; *Radoslavov et al., 2017*).

In Serbia, there were described three ecotypes (Banat, Syenichko–Peshterski and Timok), which differed in morphology and chromosome structure (*Stanimirovic, Stevanovic & Andjelkovic, 2005*). The Banat ecotype is also characterised by yellow coloration on the scutellum and anterior tergites (*Grozdanic, 1926*). The Serbian population of *A. m. carnica* was more similar to the populations in Hungary and western Romania than to the populations in Austria and northern Slovenia (*Ruttner, 1988a*, p. 255). The region of former Yugoslavia has a long tradition of beekeeping (*Kulinčević, 1959*). In this territory, honey bee breeding started in the 1980s (*Rašić, Mladenović & Stanisavljević, 2015*). Since 1990, as a result of changes in political boundaries, honey bee breeding has been carried out at the country level. In Serbia, there are six selection centres (*Hatjina et al., 2014*), and it is not allowed to breed other subspecies than *A. m. carnica* (*Official Gazette of the Republic of Serbia, 2011*). In order to preserve native Serbian bees, it is important to control whether the bees artificially selected by breeders do not differ significantly from the surrounding population, which is under natural selection.

The objective of this study is to investigate the geographical variation of honey bees in Serbia and neighbouring countries. In order to describe the bees, we used geometric morphometrics of forewing venation. We verified how the breeding of honey bees by queen producers affects the geographic variation of honey bees.

## MATERIALS AND METHODS

For this study, we have newly obtained 2,282 honey bee wing images from two datasets. The first dataset represented 1,081 workers and 56 colonies (sample numbers: 21–76) collected in 2022 and stored in alcohol. Each colony represents a different location in Serbia (Fig. 1). Most colonies in the first sample were represented by 20 workers. If possible, we have used both the left and the right wings. Forewings were carefully detached and dry-mounted between the two microscopic slides, and their images were obtained using a camera equipped with a 25 mm lens (RICOH FL-CC2514-2M). The second dataset represented 136 workers and four colonies (sample numbers: 77–80) collected in 2005. Each of the colonies in the second dataset represents a different location in Serbia (Fig. 1). The right forewings were permanently mounted on microscopic slides soon after their collection. Images of the wings were obtained using a Nikon SMZ18 microscope equipped with a Nikon DSRi2 camera. The wing images were saved in PNG files at a resolution of 94,488 pixels per meter. In the analysis, we also used the third dataset from Serbia, which was published earlier (*Rašić, Mladenović & Stanisavljević, 2015*; *Oleksa et al., 2022*, *2023*). It was collected in 2010 and consisted of 299 left wings from 20 colonies (sample numbers: 1–20). The third dataset originated from two queen breeding apiaries located in the

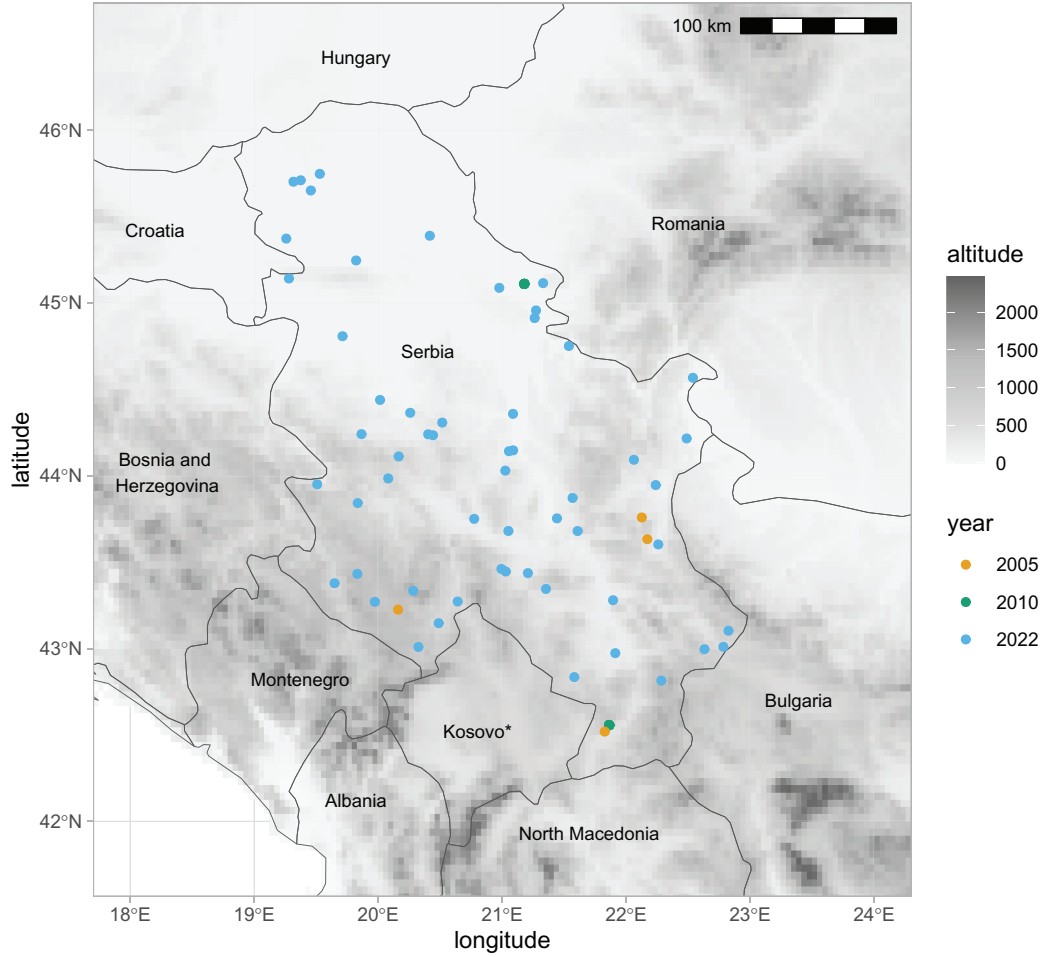

**Figure 1** **Locations in Serbia from which samples of honey bee wings were obtained in three different years.** * The designation of Kosovo is without prejudice to positions on status, and is in line with UNSCR 1244/1999 and the ICJ Opinion on the Kosovo declaration of independence. Map source: https://worldclim.org.                                                                      

north-east and south of Serbia (Fig. 1). The data from Serbia were compared with data from south-eastern Europe published in the earlier study (*Oleksa et al., 2022*). We have used the raw coordinates of 16,704 wings, representing nine nearby countries: Austria, Greece, Croatia, Hungary, Moldova, Montenegro, Romania, Slovenia, and Turkey. The third dataset from Serbia and datasets from Austria and Montenegro originated from honey bee queen breeders and are called "breeder samples". All other datasets, including the first and second datasets from Serbia, originated from beekeepers who were not involved in breeding and are called "non-breeder samples".

Nineteen landmarks were digitised on each wing using the IdentiFly software (*Nawrocka et al., 2017*; *Tofilski, 2008*). The landmarks were the same as in the earlier study (*Oleksa et al., 2023*), which allowed comparison between studies. The raw coordinates of the landmarks were saved in a CSV file and used for analysis. The newly obtained wing images and other data related to them, including landmark coordinates and geographic

coordinates of sampling locations, are available on the Zenodo website (*Kaur, Nedić & Tofilski, 2023*).

The statistical analysis was done in RStudio version 2023.09.1 using R version 4.3.2. We provide the R script for the analysis (Supplemental Document S1). The landmark coordinates were aligned using generalised Procrustes analysis from the "geomorph" package with the "gpagen" function (*Baken et al., 2021*). The aligned landmark coordinates were averaged within colonies, and the sample averages were used in later analysis. The aligned landmark coordinates of samples were analysed with principal component analysis, and the first 34 principal components were used to describe wing shape. Differences in wing shape were analysed using multivariate analysis of variance. Mahalanobis distance was used as a measure of differences in wing shape. Smaller Mahalanobis distances indicate higher similarity between groups. The association between the first two principal components and geographic coordinates was visualised using generalised additive model (GAM) regression in the mgcv package (v. 1.9-0) (*Wood, 2011*). Canonical variate analysis and leave-one-out cross-validation were calculated using the Morpho package (v. 2.11) (*Schlager, 2017*). Similarity between countries was analysed using the unweighted pair group method with arithmetic mean (UPGMA) in the phangorn package (v. 2.11.1) (*Schliep, 2011*).

## RESULTS

### Including breeder samples

The wing shape of Serbian honey bees differed significantly between years (multivariate analysis of variance: F = 3.30, $P = 7.52 \times 10^{-8}$). Principal component analysis revealed that breeder samples, collected in 2010, differed from non-breeder samples, collected in 2005 and 2022 (Fig. 2A). Spatial analysis of Serbian samples from all years revealed a complicated and unexpected pattern of geographical variation of PC1 (Fig. 3A). In pairwise comparisons, Serbian breeder samples (year 2010) differed significantly from Serbian non-breeder samples (years 2005 and 2022) and all neighbouring countries except Montenegro (Table 1). Serbian breeder samples, in most cases, differed from neighbouring countries more than Serbian non-breeder samples (Table 1). The exception was Montenegro, which, in terms of wing shape, was more similar to Serbian breeder samples than to Serbian non-breeder samples. Surprisingly, the Mahalanobis distance between the Serbian breeder samples and samples from Montenegro (2.085) was markedly smaller than the distance between Serbian breeder and non-breeder samples from 2005 and 2022 (5.746 and 5.039, respectively). On the other hand, Serbian non-breeder samples from 2005 and 2022 were similar and did not differ from each other. The UPGMA similarity tree shows that Serbian breeder samples are most similar to Montenegro, and they are clearly different from Serbian non-breeder samples and neighbouring Serbia, including Croatia, Hungary, and Romania (Fig. 4A). On the other hand, Serbian non-breeder samples were most similar to each other, and they were located in the middle of the cluster consisting of neighbouring countries (Fig. 4A).
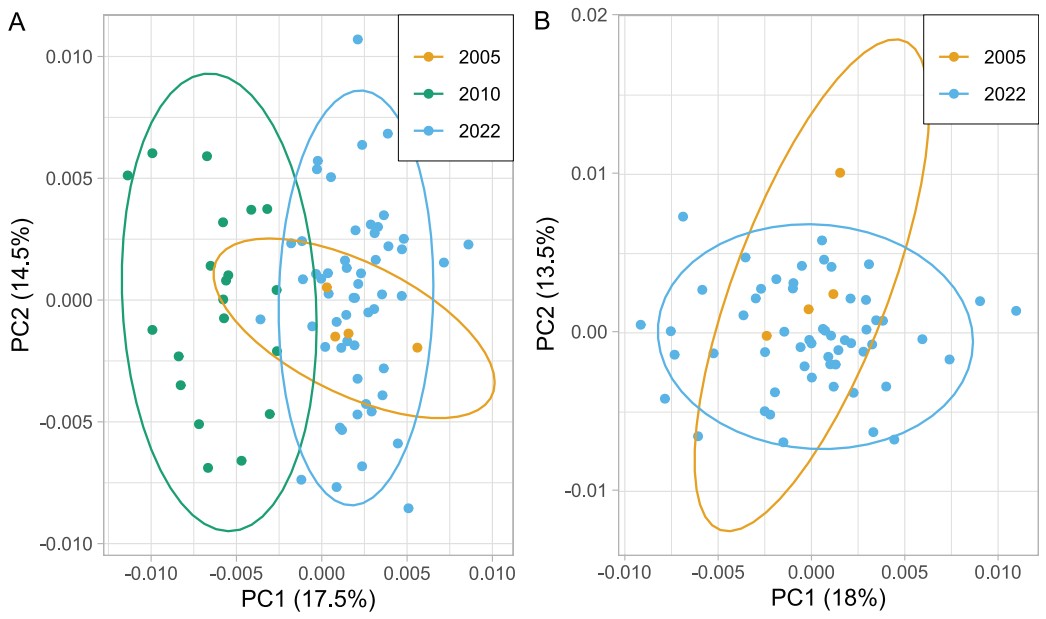

**Figure 2 Variation of the wing shape in Serbia represented by the first two principal components when breeder samples (year 2010) were included (A) or excluded (B).** Ellipses indicate 95% confidence regions.

## Excluding breeder samples

Serbian breeder samples (from 2010) were removed from further analysis because they differed markedly from non-breeder samples (from 2005 and 2022) and their geographical variation was unexpected. There were also samples removed from Austria and Montenegro because they originated from queen breeders. After the removal of the breeder samples, the remaining Serbian samples formed a single cluster in the graph of the first two principal components (Fig. 2B). The first principal component of wing shape was decreasing continuously from south-west to north-east (Fig. 3B). The geographical variation at the country level was similar to the geographical variation of wing shape at the regional level (Fig. 5).

When compared with countries from south-eastern Europe, Serbian non-breeder samples were in the centre of the graph of the first two canonical variates (Fig. 6). Serbian non-breeder samples differed significantly from all other countries (Table 2). They were most similar to Croatia (Table 2). In the similarity tree, Serbia is in the same cluster as Croatia, Slovenia, and Hungary. It is also similar to the cluster consisting of Romania and Moldova but differs more from the cluster consisting of Greece and Turkey (Fig. 4B). Most (98.3%) of the non-breeder colonies were correctly classified as Serbia, and only one colony (1.6%) was incorrectly classified as Croatia. Moreover, five colonies from Hungary, three from Greece, and one from Turkey were incorrectly classified as Serbia.

## DISCUSSION

The data presented here show that bred and non-bred bees from Serbia clearly differed from each other. This result confirms earlier reports from Australia where breeding

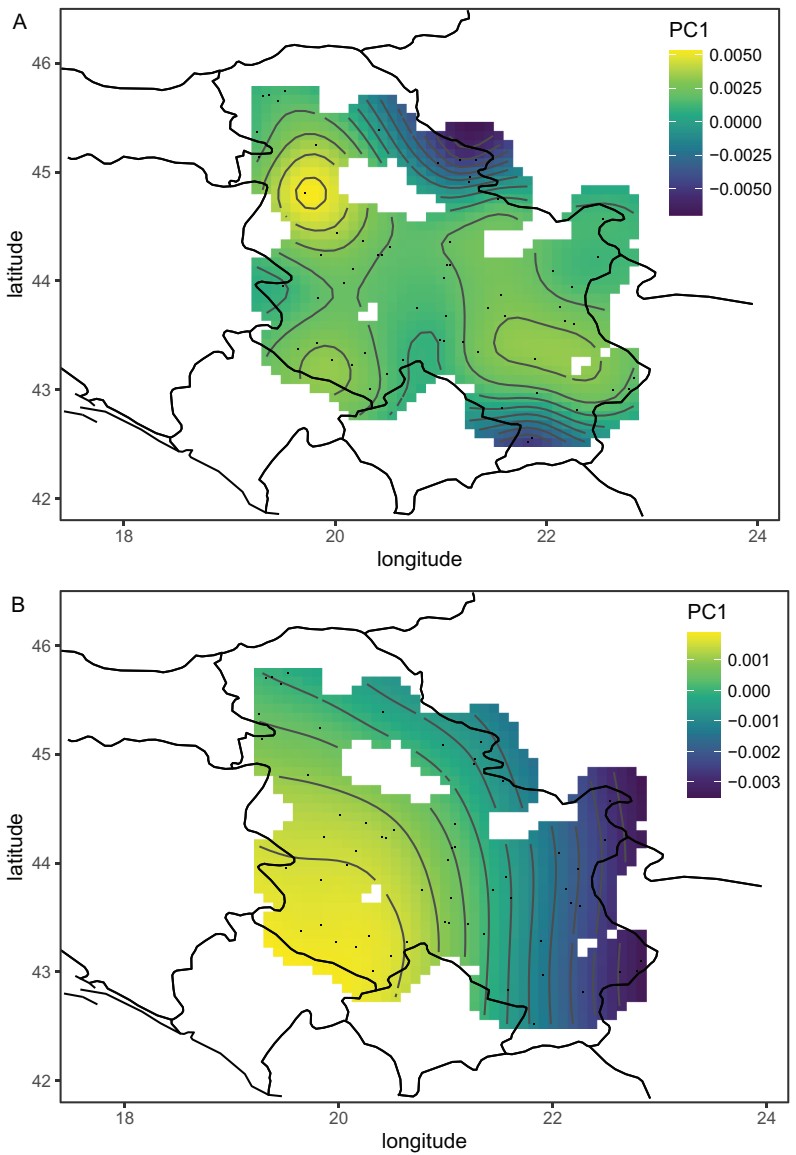

**Figure 3** The first principal component of wing shape interpolated over sampling locations using a generalised additive model when breeder samples were included (A) or excluded (B). Map source: https://worldclim.org.

populations differed from non-breeding populations, both feral and kept by beekeepers (*Chapman, Lim & Oldroyd, 2008*). At the same time, there was no difference between feral populations and those kept by beekeepers (*Chapman, Lim & Oldroyd, 2008*). A similar effect of breeding on honey bee wings was observed earlier in Austria (*Oleksa et al., 2023*) where samples from the breeding programme differed markedly from nearby populations in Slovenia, Croatia, and Hungary. The differences between breeding and non-breeding populations can be particularly high when instrumental insemination is used instead of natural mating because natural mating stations do not provide full isolation between breeding and the surrounding population (*Neumann et al., 1999*). The breeder in Austria used instrumental insemination. In consequence, the samples from Austria differed from

**Table 1** Differences in wing shape expressed as Mahalanobis distances between three datasets from Serbia and neighbouring countries (A) and probabilities of pairwise comparisons between those groups (B).

**A**

| | AT | GR | HR | HU | MD | ME | RO | RS-2005 | RS-2010 | RS-2022 | SI | TR |
|---|---|---|---|---|---|---|---|---|---|---|---|---|
| RS-2005 | 8.225 | 3.591 | 4.735 | 5.637 | 6.173 | 6.009 | 5.201 | – | 5.746 | 3.161 | 4.697 | 5.101 |
| RS-2010 | 8.036 | 5.257 | 6.682 | 6.342 | 6.597 | 2.085 | 5.269 | 5.746 | – | 5.039 | 6.469 | 7.021 |
| RS-2022 | 7.695 | 3.766 | 5.490 | 5.270 | 6.151 | 5.059 | 5.146 | 3.161 | 5.039 | – | 4.916 | 5.742 |

**B**

| | AT | GR | HR | HU | MD | ME | RO | RS-2005 | RS-2010 | RS-2022 | SI | TR |
|---|---|---|---|---|---|---|---|---|---|---|---|---|
| RS-2005 | $10^{-04}$ | 0.3399 | 0.0210 | 0.0047 | 0.0043 | 0.0014 | 0.006 | – | 0.0028 | 0.6994 | 0.0604 | $10^{-02}$ |
| RS-2010 | $10^{-04}$ | $10^{-04}$ | $10^{-04}$ | $10^{-04}$ | $10^{-04}$ | 0.5215 | $10^{-04}$ | 0.0028 | – | $10^{-04}$ | $10^{-04}$ | $10^{-04}$ |
| RS-2022 | $10^{-04}$ | $10^{-04}$ | $10^{-04}$ | $10^{-04}$ | $10^{-04}$ | $10^{-04}$ | $10^{-04}$ | 0.6994 | $10^{-04}$ | – | $10^{-04}$ | $10^{-04}$ |

**Note:**
Abbreviations: AT, Austria; GR, Greece; HR, Croatia; HU, Hungary; MD, Moldova; ME, Montenegro; RO, Romania; RS, Serbia; SI, Slovenia; TR, Turkey.

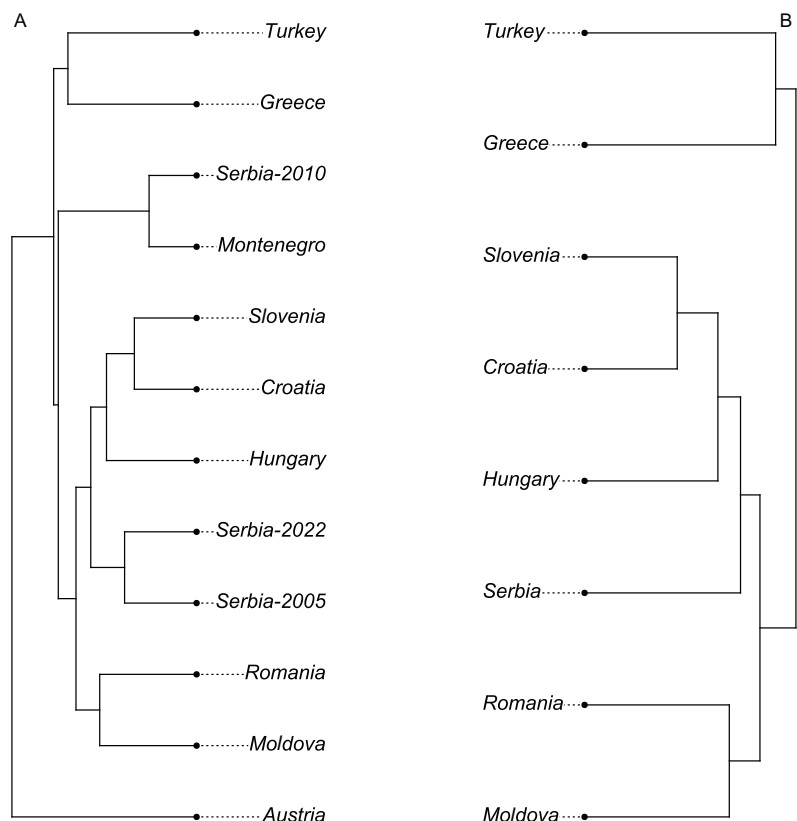

**Figure 4** UPGMA tree showing similarity between countries in wing shape when breeder samples were included (A) or excluded (B).

all other samples in lineage C (Fig. 4A). In the case of breeders from Serbia and Montenegro, natural insemination was used (*Oleksa et al., 2023*); therefore, the differences with neighbouring non-breeding populations were smaller.
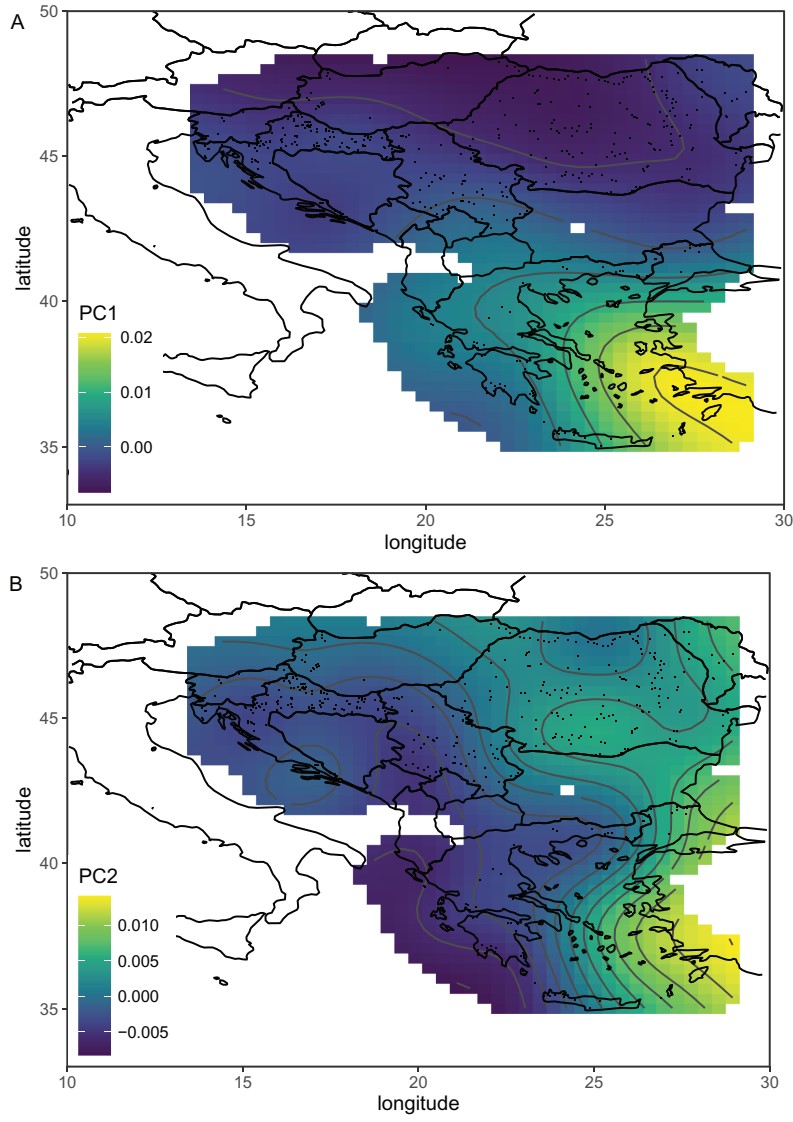

**Figure 5 The first (A) and second (B) principal components of wing shape interpolated over sampling locations in south-eastern Europe using a generalised additive model.** Breeder samples were excluded from this analysis. Map source: https://worldclim.org.

It is interesting that the Serbian breeding population became morphologically distinct not only from the Serbian non-breeding population but also from all neighbouring non-breeding populations. Differences in wing venation between countries, measured as Mahalanobis distance, were in most cases higher for breeding populations than non-breeding populations (Table 1A). The notable exception was Montenegro, which was more similar to Serbian breeding than the non-breeding population. Surprisingly, Serbian breeding samples were more similar to those from Montenegro (MD = 2.085) than to Serbian non-breeding samples from 2005 and 2022 (MD = 5.746 and 5.039 respectively). This unexpectedly high similarity can be explained by the exchange of breeding material between breeders who are in the same breeding programme.

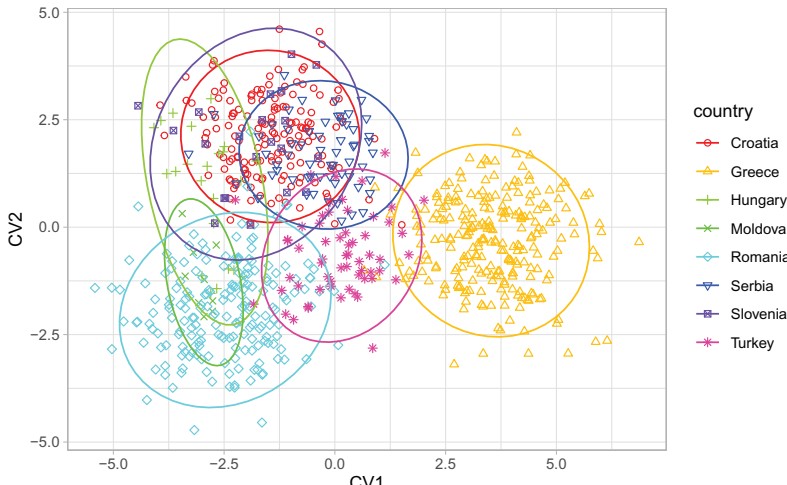

**Figure 6 Discrimination between countries based on the first two canonical variates.** Ellipses indicate 95% confidence regions. Breeder samples were excluded from this analysis.

**Table 2 Differences in wing shape expressed as Mahalanobis distances between countries (lower triangle) and significance of differences between countries (upper triangle).** Breeder samples were excluded from this analysis.

|          | Croatia | Greece | Hungary | Moldova | Romania | Serbia | Slovenia | Turkey |
|----------|---------|--------|---------|---------|---------|--------|----------|--------|
| Croatia  | –       | $10^{-04}$ | $10^{-04}$ | $10^{-04}$ | $10^{-04}$ | $10^{-04}$ | $6 \times 10^{-04}$ | $10^{-04}$ |
| Greece   | 5.615   | –      | $10^{-04}$ | $10^{-04}$ | $10^{-04}$ | $10^{-04}$ | $10^{-04}$ | $10^{-04}$ |
| Hungary  | 3.554   | 7.175  | –       | $10^{-04}$ | $10^{-04}$ | $10^{-04}$ | $10^{-04}$ | $10^{-04}$ |
| Moldova  | 5.662   | 7.689  | 5.958   | –       | $10^{-04}$ | $10^{-04}$ | $10^{-04}$ | $10^{-04}$ |
| Romania  | 4.225   | 6.234  | 4.157   | 4.249   | –       | $10^{-04}$ | $10^{-04}$ | $10^{-04}$ |
| Serbia   | 3.675   | 5.312  | 5.186   | 6.033   | 5.029   | –      | $10^{-04}$ | $10^{-04}$ |
| Slovenia | 2.720   | 6.390  | 4.275   | 5.427   | 4.704   | 4.876  | –        | $10^{-04}$ |
| Turkey   | 5.197   | 5.610  | 6.196   | 7.463   | 5.323   | 5.555  | 6.322    | –      |

In order to preserve native bees, breeders should stop importing non-native genetic material and include in the breeding programme a relatively large number of colonies from the native population. High genetic variability should be maintained by occasional screening of the non-breeding population in search of suitable colonies that can be included in the breeding population. Queens should be naturally inseminated in mating stations surrounded by protected areas where the importation of non-native bees is prohibited. Apiaries within the protected areas should be occasionally tested to detect illegal imports. Under morphometric or genetic control, there should be all colonies in the breeding programme, and those that differ significantly from the native population should be excluded. There are potential trade-offs between preservation of native genetic variability and fast selection progress; however, maintaining ecological adaptability and resilience should be considered more important than productivity because a stable honey

bee population providing pollination services (*Büchler et al., 2014*) is more important than a potential increase in honey production (*Bienefeld, 2016*).

In an earlier study about the geographical variation of honey bees in Romania, the differences between years were explained by temporal changes related to introgression (*Tofilski et al., 2021*). However, in the present study, breeding samples from 2010 differed from both earlier (2005) and later (2022) samples. On the other hand, Serbian non-breeding samples did not differ significantly from each other despite the relatively long time between their collection in 2005 and 2022. Moreover, the wing shape of the breeding samples from Serbia differed markedly from neighbouring non-breeding populations.

Although the artificial selection used during breeding is mainly focused on traits important for beekeeping, including honey production, defensive behaviour, or disease resistance (*Büchler et al., 2013*; *Hoppe et al., 2020*), breeding lines are often assigned to particular subspecies, and breeders attempt to maintain their morphological similarity to the focal subspecies (*Lodesani & Costa, 2003*). This is usually achieved by measurements of the cubital index (*Ruttner, 1988b*) or other morphological traits (*Lodesani & Costa, 2003*). Recently, genetic markers have also been used more often for this purpose (*Oldroyd & Thompson, 2006*). The morphological differences between breeder and non-breeder populations can be related not only to selection but also to genetic drift. Especially in the case of closed population breeding, the population size is often relatively small, and there is a random loss of genetic variation (*Page & Laidlaw, 1982*).

Including breeding samples in the analysis of geographic variation of wing venation in Serbia resulted in unexpected and complicated results (Fig. 3A). The variation was completely different when only non-breeding samples were used for this analysis (Fig. 3B). We believe that the analysis without breeder samples better represents the natural situation because it agrees with similar analyses on a larger scale (Fig. 5). The variation originating from breeding is not interesting in the context of geographical variation because it is not a result of natural selection; therefore, breeder samples should not be included in such analysis.

The samples most unsuitable for analysis of geographic variation are those obtained from colonies headed by queens that were instrumentally inseminated. Fortunately, most of the queens distributed by the queen breeder are uninseminated, and they mate naturally with local drones. In this case, the influence of the introduced queen is smaller. In studies of the geographic variation of honey bees, samples should be collected only from colonies of local origin that are headed by naturally mated queens, not from breeders. The risk of collecting bees affected by breeding is lower if the samples are collected from flowers because, in most locations, only a fraction of colonies are headed by introduced queens. The samples from flowers should be collected at a distance from any apiary to avoid the risk that most workers in the sample originate from a single beekeeper who can use introduced queens. If only a fraction of the analysed bees were affected by breeding, the natural pattern of geographical variation could still be detected. In the analysis covering a large part of south-eastern Europe, the geographic variation of wing venation (Fig. 5) was similar to results based on samples obtained from both breeders and non-breeders (Fig. 4

in *Oleksa et al., 2023*), because the fraction of samples originating from breeders was relatively small.

The first principal component, which described a large part of the variation, formed a gradient in south-eastern Europe from the south-east to the north-west (Fig. 5A). This gradient represents a transition from lineage O, which is present in Asiatic parts of Turkey and some Greek islands, to lineage C, which is present in most of the study area. The second principal component formed a more complicated pattern, but it largely varied from east to west (Fig. 5B). It needs to be stressed that the Figs. 5A and 5B are in some parts interpolated, and more data is required from Bosnia and Herzegovina, Montenegro, Albania, North Macedonia, and Bulgaria. After the removal of breeder samples, there is visible in Serbia a pattern that is consistent with large-scale geographic variation in south-eastern Europe (Fig. 5B). The geographic variation of wing shape presented here agrees to some degree with an earlier study based on mitochondrial DNA and microsatellites (*Muñoz & De la Rúa, 2021*), where there is a succession of ecotypes from south-east to north-west: cecropia, macedonica-2, macedonica-1, carnica-2, and carnica-1. A relationship between geographic coordinates and haplotype diversity was also reported in another study based on mitochondrial DNA (*Tanasković et al., 2021*).

The data presented here do not agree very well with the subspecific partitioning of south-eastern Europe proposed by *Ruttner (1988a)*, which assumed the presence of three honey bee subspecies: *A. m. carnica*, *A. m. cecropia*, and *A. m. macedonica*. A. m. carnica should occur in south-eastern Austria, Slovenia, Croatia, Hungary, north-western Romania, Bosnia and Herzegovina, most of Serbia, Montenegro, and Albania. *A. m. cecropia* should occur in southern and central Greece, and *A. m. macedonica* should occur in the northern part of Greece, North Macedonia, the European part of Turkey, Bulgaria, south-eastern Serbia, south-eastern Romania, and Moldova. In *Ruttner*'s *(1988a)* approach, *A. m. carpatica* from Romania (*Foti et al., 1965*) and *A. m. rodopica* from Bulgaria (*Petrov, 1995*) were not included. The data presented here do not confirm any clear boundaries between the subspecies proposed by *Ruttner (1988a)*. The wing venation differed markedly between countries (Table 2); however, the geographic variation was continuous and markedly different from the predicted distribution of the subspecies. For example, both Moldova and Turkey are in the proposed range of *A. m. macedonica* and should be similar to each other, but they differ markedly (MD = 7.463). On the other hand, the difference between Moldova and Slovenia is much smaller (MD = 5.427), despite the fact that they should represent two different subspecies (*A. m. macedonica* and *A. m. carnica*, respectively). The boundaries between the subspecies proposed by Ruttner largely neglect the mountains as barriers to gene flow. For example, apart from the Carpathians, there are no geographical barriers between *A. m. macedonica* and *A. m. carnica*. The geographic variation of wing venation (Figs. 5A and 5B) should reflect the presence of geographical barriers to gene flow. It is interesting that in the data presented in this study, there is a visible effect of the Carpathians (Fig. 5A). We can also predict that a similar effect will be visible in the case of other mountains. For example, there are mountain ranges in south-western Bulgaria that should be important barriers to gene flow between Greece and Bulgaria. Another mountain range stretches from Bosnia and Herzegovina to Greece,
which should isolate the Adriatic coast from the inland. Unfortunately, gaps in coverage of the dataset presented here do not allow us to verify how those mountains affect variation in wing venation.

## CONCLUSIONS

Honey bees under artificial selection differ in wing venation from surrounding populations, which are under natural selection.

Samples obtained from queen breeders should not be included in the analysis of the natural geographical variation of honey bees.

In south-eastern Europe, there is continuous variation in wing venation, and, apart from the Carpathians, there is no clear boundary between *A. m. carnica*, *A. m. cecropia*, and *A. m. macedonica*.

## ACKNOWLEDGEMENTS

We express our gratitude to the beekeepers who kindly contributed samples of honey bees. We also thank anonymous reviewers for their constructive feedback and suggestions.

### Funding

This research was funded by National Science Centre, Poland, grant number: 2021/41/B/NZ9/03153. The funders had no role in study design, data collection and analysis, decision to publish, or preparation of the manuscript.

### Grant Disclosures

The following grant information was disclosed by the authors:
National Science Centre: 2021/41/B/NZ9/03153.

### Competing Interests

The authors declare that they have no competing interests.

### Author Contributions

- Hardeep Kaur performed the experiments, prepared figures and/or tables, authored or reviewed drafts of the article, and approved the final draft.
- Nebojša Nedić performed the experiments, authored or reviewed drafts of the article, and approved the final draft.
- Adam Tofilski conceived and designed the experiments, analyzed the data, prepared figures and/or tables, authored or reviewed drafts of the article, and approved the final draft.

### Data Availability

The newly obtained wing images and other data related to them, including landmark coordinates and geographic coordinates of sampling locations, are available at Zenodo:

Kaur, H., Nedić, N., & Tofilski, A. (2023). Fore wing images of honey bees (*Apis mellifera*) from Serbia [Data set]. Zenodo. https://doi.org/10.5281/zenodo.10389960

## Supplemental Information

Supplemental information for this article can be found online at http://dx.doi.org/10.7717/peerj.17247#supplemental-information.

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
