# Peer review of "Influence of honey bee (Apis mellifera) breeding on wing venation in Serbia and neighbouring countries"

_PeerJ, doi:10.7717/peerj.17247_

## Round 0.1 · original submission · Minor Revisions

Dear author,

Although your study is robust, supported by successful statistical analyses which provide a solid foundation for conclusions drawn, and the manuscript is well written, there are some minor revisions suggested by the reviewers. Corrections suggested by reviewers 2 & 3 should be considered.

Reviewer 1 ·

Basic reporting

Dear authors I have some comments of your manuscript:
Introduction:
Line number 34 (Uzunov, Brascamp & B¸chler, 2017) correct to Uzunov et al., 2017
Line number 59 &60 (Oleksa, 60 Kusza & Tofilski, 2021) correct to Oleksa et al., 2021
Line number 67 & 68 (Cobey, Sheppard & 68 Tarpy, 2012) correct to Cobey et al., 2012
Line number 87 (Ivanova, Bienkowska & Petrov, 2011). Correct to Ivanova et al., 2011
Line number 90 (Stanimirovic, Stevanovic & Andjelkovic, 2005) correct to Stanimirovic et al., 2005
Line number 95 &96 (Rasic, Mladenovic & Stanisavljevic, 2015). Correct to Rasic et al., 2015
Materials and Methods
Line number 117 Rasic, Mladenovic & Stanisavljevic, 2015). Correct to Rasic et al., 2015
Line number 129 (Kaur, Nedic & Tofilski, 2023). Correct to Kaur et al., 2023
Results
What did you mean by Serbian non-breeder samples? Also in the materials and methods you should be explain the differences between Serbian breeder samples and Serbian non- breeder samples.
Discussion
Line number 213 What did you mean by neighbouring non-breeding populations
Line number 269 (1988a). correct to Ruttner 1988 a

Experimental design

no comment

Validity of the findings

no comment

Reviewer 2 ·

Basic reporting

Although English is not my native language, I believe the manuscript is well-written and is very clear in its statements. The literature used is updated, and the authors explore it cleverly, citing the most important papers dealing with the subject analyzed in the manuscript. The hypothesis is clear and well structured and the results are presented with figures of good quality, easy to interpret, and well explored along the manuscript. I have only one small comment regarding the manuscript's readability, especially in the first two paragraphs. All the sentences are extremely short, making the text's readability difficult. If possible, I suggest the authors reorganize the manuscript by fusing sentences with continuous ideas to make it easier for the reader.

Experimental design

The experimental design is well-planned, the sample sizes are adequate, and the used approaches are indicated to answer the questions raised by the hypothesis. Although the results are more or less expected for this kind of question, the manuscript brings an approach using geometric morphometrics that is an important tool for many morphological evaluations, such as the one in question in this manuscript. Thus, I think the manuscript is welcome to be published in a journal like PeerJ, as it meets the Aims and Scope of the Journal.

Validity of the findings

The obtained data are robust and bring an important confirmation to the field of Genetic Variability of managed populations of Apis mellifera. The results are well-interpreted, and the conclusions based on these results are plausible and replicable. The discussion is well-delimited and based on recent literature, and none of the drawled conclusions are out of the scope of the data or the previously established hypothesis. So, in my opinion, I would have no further comments or suggestions for the authors other than the readability of the first paragraphs of the manuscript.

Reviewer 3 ·

Basic reporting

Language and Professionalism:
The manuscript is written in clear, professional English, making it accessible to an international audience. The authors have done a good job in communicating their research findings in a coherent and unambiguous manner.

Introduction & Background:
The background provided is comprehensive, effectively setting the context for the study. The literature is well-referenced, demonstrating the relevance of this work within the broader field of honey bee research.

Structure:
The manuscript conforms to the expected standards, with a logical flow from introduction to conclusions. The use of figures is appropriate, enhancing the clarity of the presented data. Raw data availability on Zenodo is commendable, aligning with PeerJ's policy for transparency and reproducibility.

Experimental design

Scope and Originality:
The study falls well within the scope of the journal, offering original insights into the impacts of artificial selection on honey bee wing venation and its implications for geographic variation analysis.

Research Question and Gap:
The research question is defined, addressing a gap in understanding the effect of breeding on the natural geographic variation of honey bees. The rationale behind the study is compelling, emphasizing the importance of distinguishing between artificially selected and natural populations in morphometric analyses.

Methodological Rigor:
The experimental design is solid, with a clear description of the methodology. The inclusion of a diverse dataset and the use of geometric morphometrics are strengths of this study.

Validity of the findings

Data Robustness and Statistical Analysis:
The data presented are robust, supported by statistical analyses. The use of principal component analysis, multivariate analysis of variance, and other statistical methods provides a solid foundation for the conclusions drawn.

Conclusions:
The conclusions are well-supported by the results, directly tying back to the original research question. The findings contribute insights into the field, particularly regarding the implications of breeding on the analysis of geographic variation.

Additional comments

This study is a contribution to the understanding of honey bee diversity and the impact of artificial selection. The methodology is solid, and the data analysis is thorough. One suggestion for improvement would be to further discuss the implications of these findings for conservation efforts and breeding practices.
I have some minor suggestions:
Line 89-99: If this paragraph is important, then please make it more clear to the reader why.
The study primarily focuses on the morphological differences between bred and non-bred populations without delving deeply into the broader implications of these findings for conservation strategies. An expanded discussion on how breeding practices might be optimized to preserve genetic diversity and support conservation efforts would strengthen the paper.
The manuscript could benefit from a more detailed examination of the selection criteria used by breeders and how these criteria align or diverge from the traits valued for conservation or ecosystem services. This would offer insights into the potential trade-offs between breeding for productivity and maintaining ecological adaptability and resilience.
The paper discusses instrumental insemination but could further explore its implications for genetic diversity and population structure and highlight the many downsides of this technique.
The study captures data from different years but could further investigate the temporal dynamics of wing venation changes, considering factors such as climate change, habitat alteration, and shifts in beekeeping practices over time.

---

## Round 0.2 · accepted · Accept

Dear Dr. Tofilski,

Thank you for addressing all of the reviewers' comments. The manuscript conforms to the expected standards, with a logical flow from introduction to conclusions. The use of figures is appropriate, enhancing the clarity of the presented data.